# Structural and functional analysis of target recognition by the lymphocyte adaptor protein LNK

Rhiannon Morris [1,2], Yaoyuan Zhang[3,4], Julia I. Ellyard [3,4], Carola G. Vinuesa [3,4], James M. Murphy [1,2], Artem Laktyushin[1,2], Nadia J. Kershaw [1,2 ✉] & Jeffrey J. Babon [1,2 ✉]

The SH2B family of adaptor proteins, SH2-B, APS, and LNK are key modulators of cellular signalling pathways. Whilst SH2-B and APS have been partially structurally and biochemically characterised, to date there has been no such characterisation of LNK. Here we present two crystal structures of the LNK substrate recognition domain, the SH2 domain, bound to phosphorylated motifs from JAK2 and EPOR, and biochemically define the basis for target recognition. The LNK SH2 domain adopts a canonical SH2 domain fold with an additional N-terminal helix. Targeted analysis of binding to phosphosites in signalling pathways indicated that specificity is conferred by amino acids one- and three-residues downstream of the phosphotyrosine. Several mutations in LNK showed impaired target binding in vitro and a reduced ability to inhibit signalling, allowing an understanding of the molecular basis of LNK dysfunction in variants identified in patients with myeloproliferative disease.

[1] Walter and Eliza Hall Institute of Medical Research, 1G Royal Parade, Parkville, VIC 3052, Australia. [2] Department of Medical Biology, The University of Melbourne, Royal Parade, Parkville, VIC 3052, Australia. [3] Australia Department of Immunology and Infectious Diseases, Australian National University, Canberra, ACT, Australia. [4] Australia Centre for Personalised Immunology, John Curtin School of Medical Research, Australian National University, Canberra, ACT, Australia. ✉email: kershaw@wehi.edu.au; babon@wehi.edu.au

   

Thrombopoietin (TPO) and erythropoietin (EPO) are hematopoietic cytokines that bind specific receptors on the surface of target cells and induce activation of the Janus kinase (JAK)-signal transducer and activation of transcription (STAT) pathway. TPO is an essential modulator of mega-karyopoiesis, platelet production and stem cell quiescence, whereas EPO is required for the regulation of erythropoiesis. The Lymphocyte adaptor protein (LNK[1] (also SH2B3)) is a member of the SH2 domain containing adaptor family of proteins, which also comprises APS (SH2B2) and SH2B (SH2B1)[2,3] and negatively regulates both EPO and TPO signalling[4,5] via its interaction with JAK2[6]. All three proteins contain a dimerisation domain, a pleckstrin homology (PH) domain and a Src homology 2 (SH2) domain and are involved in the regulation of various signalling pathways downstream of cytokines and growth factors[4,5,7–10].

LNK is highly expressed in haematopoietic stem cells (HSCs), and deletion of LNK from HSCs leads to an increase in cell number and proliferative capacity[6], suggesting a role for LNK in regulating HSC self-renewal. Comparably, overexpression of LNK in haematopoietic progenitor cell lines restrains TPO-induced cellular proliferation, and overexpression of LNK in primary hematopoietic cells inhibits megakaryopoiesis[4]. These findings are recapitulated in vivo with LNK-deficient mice displaying increased numbers of megakaryocytes that have enhanced TPO sensitivity[4] as well as enhanced numbers of platelets, lymphocytes and erythroid cells[3,5]. Similarly, erythroid colony-forming progenitors from the spleens of LNK deficient mice displayed increased sensitivity to EPO stimulation[5].

Given the ability of LNK to negatively regulate EPO and TPO signalling, it is unsurprising that LNK loss-of-function mutations have been identified as drivers of human myeloproliferative disease[11]. In addition, there is an increased incidence of LNK mutations in leukaemic transformation of myeloproliferative neoplasms, suggesting LNK function may effect the severity of disease[12–14]. The PH and SH2 domains of LNK are both hotspots for mutations[15], and are essential for LNK function. Currently the mechanism by which LNK negatively regulates signalling is poorly understood, and so the role of LNK substitutions in the onset and progression of disease is uncertain. Understanding how the LNK SH2 domain interacts with substrates may shed light on how negative regulation of various signalling molecules occurs, and why mutations in the LNK SH2 domain contribute to disease burden in patients.

LNK has also been suggested to bind a suite of other signalling proteins involved in haematopoiesis, proliferation and differentiation, including c-KIT[16], FLT3[17], c-FMS[18] and PDGFR[19]. The need to establish which signalling proteins are regulated by LNK is underscored by the identification of LNK mutations in patients with a range of inflammatory, immune and haematopoietic diseases including cancer[20–25]. Characterisation of the substrate recognition domain of LNK, the SH2 domain, would therefore be a step towards a comprehensive overview of the proteins and pathways that LNK negatively regulates. While the structural details of APS and SH2-B have been elegantly characterised by the Hubbard laboratory[2,26,27], to date there have been no structural or biochemical studies of LNK, with only a single published example of the successful expression of any part of the LNK protein[28] and no example of the yields and purity required for an in-depth structural and biochemical analysis.

Here we present the structure of the substrate recognition domain of LNK, the SH2 domain, in complex with phospho-peptide motifs from JAK2 (pY813) and EPOR (pY454) and biochemically characterise these interactions. The LNK SH2 domain, unlike APS but similarly to SH2-B, adopts a monomeric structure and binds its target sequences in a canonical linear, extended conformation. We tested the ability of the LNK SH2 domain to bind to a number of different phosphorylated peptides from proteins proposed to be regulated by LNK, revealing the highest affinities were for sites on JAK2, JAK3 and EPOR. Studies of SH2 domain point mutations identified in patients with myeloproliferative neoplasms revealed that the SH2 domain displayed a decrease in affinity for ligands and the same mutations, when incorporated into the full-length protein had a reduced capacity to regulate signalling in vitro. Together these findings detail the specificity of the LNK SH2 domain and aid our understanding of how single point mutations affects LNK function, contributing to haematological diseases.

## Results

### Crystal structure of the LNK SH2 domain in complex with the JAK2 pY813 motif.
We attempted a number of different bacterial and baculovirus expression systems to produce active SH2 domain from both human and mouse LNK including. Whilst GST and His6-based fusion systems were unsuccessful, a NusA fusion of the *M. musculus* LNK SH2 domain (using the domain boundaries from Machida et al.[28]) yielded folded, monomeric protein once the fusion tag was removed as determined by size exclusion chromatography and thermal shift assay. The LNK SH2 domain was then co-crystallised with a 12-mer phosphopeptide corresponding to JAK2 pY813 and flanking residues (designated below as +1, +2…. and −1, −2… relative to pTyr). Importantly, the murine LNK SH2 domain shares 92.9% identity with the *H. sapiens* orthologue (Supplementary Fig. 1), having high conservation across all phosphopeptide-binding residues, with only a threonine to serine substitution from human to mouse within the phosphotyrosine binding pocket. The crystal structure of this complex was solved to 1.9 Å resolution (Supplementary Table 1), revealing that the LNK SH2 domain adopts a typical SH2 domain fold comprising three central β-strands flanked by two α-helices. In addition, there is a short helix similar to that found in the SOCS and STAT proteins located just outside the SH2 domain boundary on the N-terminal side[29,30] (Fig. 1a and 1b). This helix extends behind the central β sheet and is positioned over a hydrophobic patch. A leucine in this helix (Leu330) begins an LSxYP motif conserved across all three SH2B family members and inserts its sidechain into a hydrophobic pocket on this surface (Fig. 1b and Supplementary Fig. 2). Thus, this N-terminal helix may be a conserved feature across the SH2B family. Additionally, this helix appears to be conserved across several vertebrate species (Supplementary Fig. 2), suggesting this N-terminal helix may be an evolutionarily conserved feature in LNK. Another unexpected feature is a disulfide bond formed between Cys 421 and Cys 425 of the BG loop, a feature not present in SH2B or APS; whether this would be present in vivo is unclear.

The pTyr of the JAK2 pY813 phosphopeptide inserts into the canonical phosphotyrosine binding pocket of the LNK SH2 domain, formed by Arg 343, Arg 364, Ser 366, Arg 369, His 385 and Arg 387. The phosphate moiety of pY813 forms hydrogen bonds with the invariant Arg 364 and the highly conserved Arg 343 along with the backbone amide nitrogen of Glu 367, and the sidechain of Ser 368 (Fig. 1c, d). Glu 814 of the JAK2 peptide (P + 1) appears to be a key determinant of binding as it forms a hydrogen bond with Lys 384 of LNK. In addition, Leu 816 (P + 3) inserts into a hydrophobic pocket common in SH2 domains that is formed by Leu 386, Val 398, Leu 401, Phe 403, Phe 413, Ile 418 and Leu 420 of LNK (Fig. 1c). Numerous backbone interactions form the remainder of intermolecular contacts (Fig. 1b and Supplementary Fig. 3).

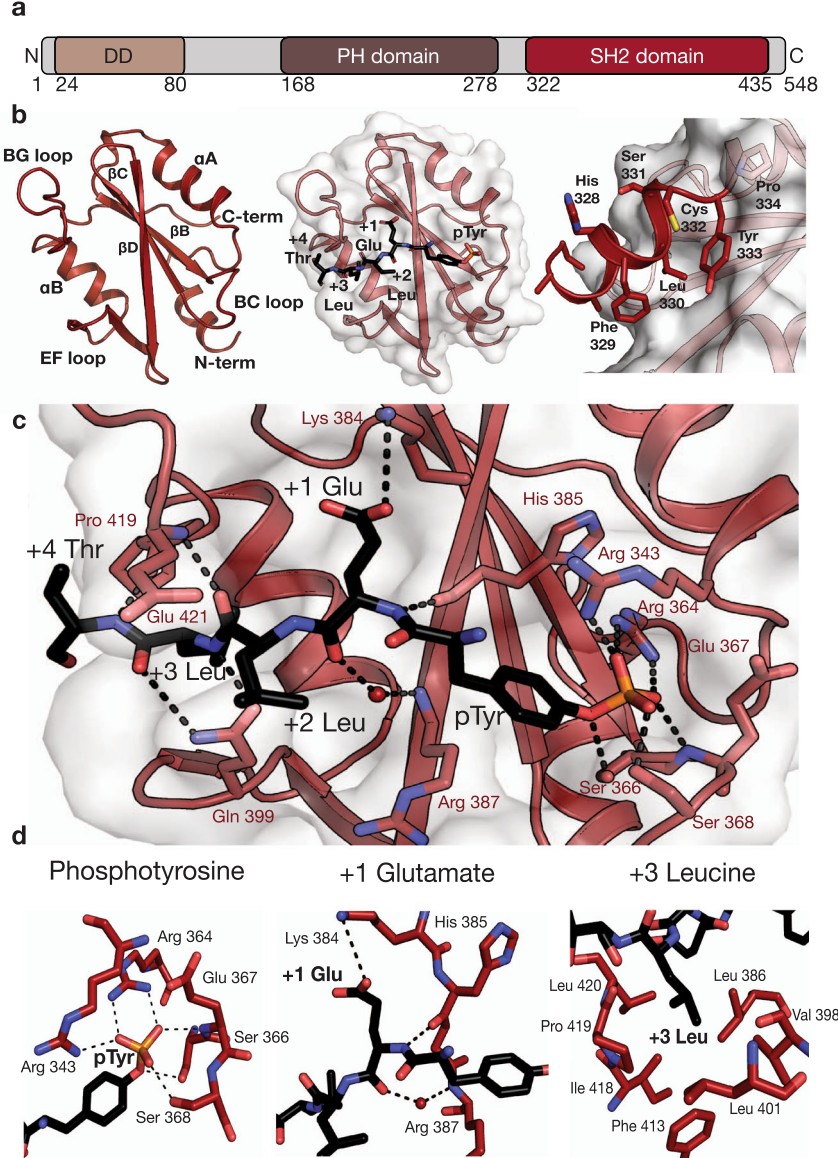

**Fig. 1 The LNK SH2/JAK2 pY813 co-crystal structure. a** Schematic representation of the LNK domain architecture. LNK comprises an N-terminal dimerisation domain, a central plekstrin homology domain and a C-terminal SH2 domain. **b** (left) Cartoon representation of the backbone of the WT LNK SH2 domain with the secondary structural features indicated (peptide not shown), (middle) LNK SH2 domain structure with the JAK2 pY813 peptide shown, and (right) the LNK SH2 domain N-terminal helix extending behind the central β strands, positioned over a hydrophobic surface on the SH2 domain. **c** Interactions between the LNK SH2 domain (red) and JAK2 pY813 peptide (black). The phosphotyrosine occupies the phosphotyrosine binding pocket of the SH2 domain and is coordinated by R364, S368, R343, and the nitrogen of E367 of LNK. The +1 Glu forms a salt bridge with K384 of LNK and the +3 Leu sits in a hydrophobic pocket formed in part by the EF and BG loops of LNK. **d** Specific interactions between the LNK SH2 domain and the phosphotyrosine, +1 Glu, and +3 Leu residues highlighted.

**Binding of the LNK SH2 domain to phosphotyrosine motifs resembles the binding mode of SH2B and is distinct from that of APS**. The SH2 domains of APS and LNK share 64.6% sequence identity (Fig. 2a), with high conservation around the phosphotyrosine binding pocket (Supplementary Fig. 4a) however the structures are notably different. APS forms a dimer (PDB ID: 1RQQ)[26] whereas LNK is monomeric. APS dimerisation occurs via its αB helix, which is longer and extends behind the SH2 domain (Fig. 2b). The +3 hydrophobic pocket forms part of this dimer interface and as such peptide binding is considerably altered (Supplementary Fig. 4a) and differs from most SH2 domains by not adopting a linear, extended conformation.

In contrast, LNK is structurally more similar SH2B, which also binds JAK2 pY813. The SH2 domains of LNK and SH2B share

68.7% sequence identity (Fig. 2a) and the crystal structures of their SH2 domains (SH2B PDB ID: 2HDX)[2] align with a root-mean-square deviation (RMSD) of 1.26 Å over 106 atoms (Fig. 2b). There are subtle differences in pTyr binding between these two family members despite a high degree of conservation across the peptide binding residues (Supplementary Fig. 4b). In both, the phosphate group of pY813 occupies the canonical phosphotyrosine binding pocket and is coordinated by an invariant arginine (Arg 364 in LNK), however in LNK, Arg 369 and 387 are involved in a hydrogen bonding network with Glu 372, and do not interact with the phosphate moiety. In contrast, in SH2B, the corresponding three residues form different interactions and the Arg 369 equivalent (Arg 560) forms a salt bridge with the phosphate of pTyr. LNK, like SH2B also displays

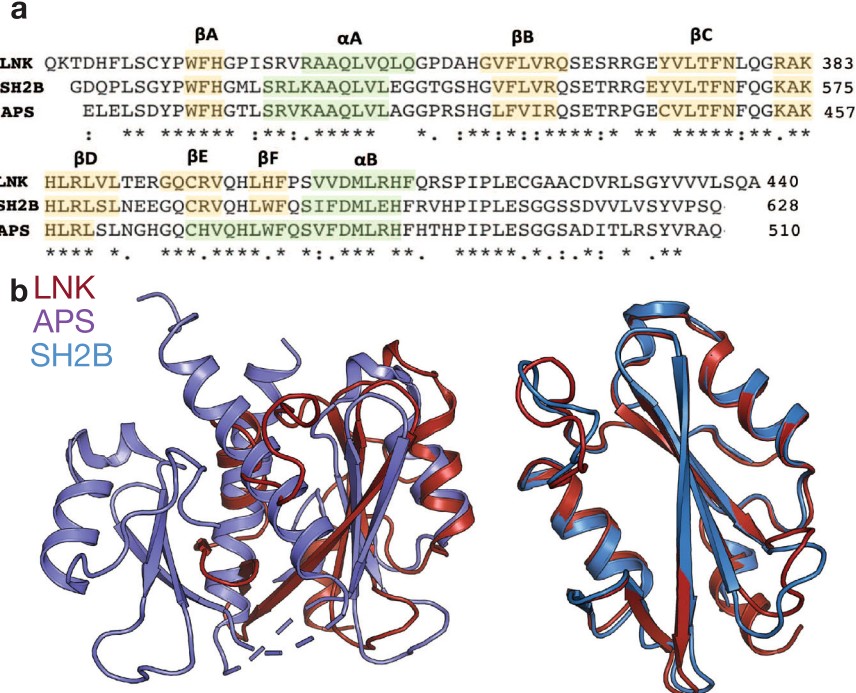

**Fig. 2 Comparison of the LNK, APS and SH2B SH2 domains. a** Alignment of APS, SH2B and LNK SH2 domain protein sequences, highlighting high sequence conservation between the three family members. **b** Alignment of the LNK and APS (PDB ID: 1RQQ) SH2 domains, with an overall RMSD of 2.4 Å and alignment of LNK and SH2B (PDB ID: 2HDX) SH2 domains with an overall RMSD of 1.3 Å.

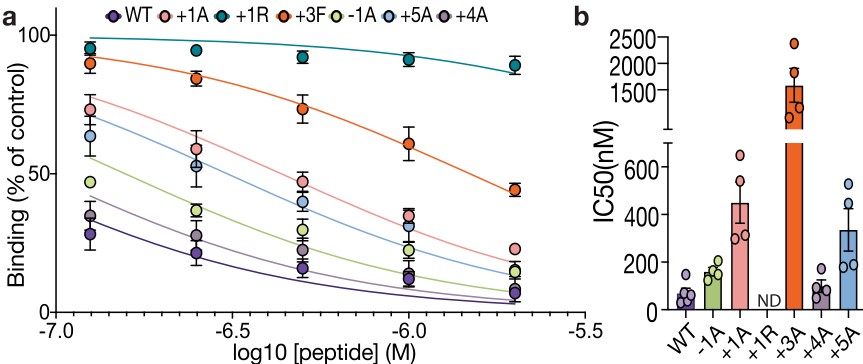

**Fig. 3 Mutational Analysis of the LNK binding site in JAK2. a** Percentage of binding (measured by the response at equilibrium) to the IL6ST pY757 peptide chip after pre-incubation with varying concentrations of mutant JAK pY813 peptides. **b** Point at which 50% of binding is inhibited shown as $IC_{50}$. For both, data are shown as mean ± SEM of technical replicates from at least two independent experiments (WT data were from five samples from $n = 3$ independent experiments, all mutant data were from four samples from $n = 2$ independent experiments).

specificity for a Glu at the +1 position, however in LNK this is facilitated by a salt bridge to Lys 384, whereas in SH2B the equivalent lysine hydrogen bonds to Ser 613 which likewise hydrogen bonds to the +1 Glu.

**Specificity for the LNK SH2 domain is conferred by amino acids one-residue, three-residues and five-residues downstream of the target phosphotyrosine.** SH2 domains typically bind ligands with a specific preference for amino acids at the −2 to +5 positions, relative to phosphotyrosine[31]. The LNK SH2/JAK2 pY813 structure suggests that +1 and +3 residues contribute to high affinity binding. To determine the contribution of individual residues, we made a series of mutations within the JAK2 phosphopeptide, and measured their affinity for the LNK SH2 domain via a surface plasmon resonance (SPR) competition assay (Fig. 3). The LNK SH2 domain was incubated with increasing concentrations of mutant JAK2 pY813 peptides before being passed

over a chip coated with an immobilised low affinity phosphopeptide derived from a site on the intracellular domain of the IL-6 receptor, IL6ST. While the $IC_{50}$ of the WT pY813 peptide was 60 nM, mutation of residues at the +1, +3, and to a lesser extent +5, positions decreased the affinity of the interaction (Fig. 3). In particular, mutation of the +1 Glu to Arg (+1R) and the +3 Leu to Phe (+3F) resulted in a >80-fold and ~25-fold decrease in affinity respectively (Table 1). The +2 residue was not mutated as the structure indicates its sidechain points towards solvent. These findings, in combination with the SH2 domain structure, indicate three residues as key binding determinants; Glu 814, Leu 816 and Glu 818, in addition to pTyr 813.

**Identification of phosphotyrosine binding motifs in JAK3, EPOR, FLT3 and c-KIT.** The LNK SH2 domain has been proposed to interact with a suite of phosphorylated sites on various signalling proteins[16–19,32]. Here, using an SPR competition assay,

**Table 1 Peptide sequences and IC$_{50}$ values from mutant JAK2 pY813 competition assay.**

| Mutant | Residues | | | | | | | IC$_{50}$ (nM) ± SEM[a] |
| --- | --- | --- | --- | --- | --- | --- | --- | --- |
| | −1 | **pY** | +1 | +2 | +3 | +4 | +5 | |
| WT | D | **pY** | E | L | L | T | E | 60 ± 20 |
| −1A | **A** | **pY** | E | L | L | T | E | 150 ± 20 |
| +1A | D | **pY** | **A** | L | L | T | E | 430 ± 90 |
| +1R | D | **pY** | **R** | L | L | T | E | >5000 |
| +3F | D | **pY** | E | L | **F** | T | E | 1430 ± 300 |
| +4A | D | **pY** | E | L | L | **A** | E | 90 ± 25 |
| +5A | D | **pY** | E | L | L | T | **A** | 300 ± 90 |

Target phosphotyrosine and mutated residues are highlighted in bold. Data are shown as mean ± SEM of technical replicates from at least two independent experiments (WT, five samples from n = 3 independent experiments, all mutants, four samples from n = 2 independent experiments).
aIC$_{50}$ as determined by SPR competition assay.

we examined binding of the LNK SH2 domain to those suggested sites from c-KIT, c-FMS, FLT3 and PDGFR and to all intracellular sites from EPOR and TPOR. In addition, binding to the activation loops of JAK2 and IRK, and pY785 of JAK3 were also examined. Although biophysical methods do not prove that an interaction exists in vivo, they provide a powerful tool to show which interactions do not occur. Given that SH2 domains will bind to most phosphorylated sequences to some degree, in line with the literature on SH2 domains and their targets[33] we applied a threshold cut-off of 2000 nM, where an IC$_{50}$ over this value indicates a physiologically irrelevant interaction. As shown in Table 2, we identified two high-affinity phosphotyrosine motifs from JAK3 and EPOR in addition to several moderate-affinity motifs from EPOR, FLT3 and c-KIT. These sequences all contain a hydrophobic residue in the +3 position. Peptides with an IC$_{50}$ greater than 2000 nM generally either lacked a hydrophobic residue at this position or contained an Arg at the +1 which we had previously shown abolished binding (Fig. 3). Despite previous reports implicating pTyr residues of PDGFR and c-FMS, as LNK ligands, in our hands, we did not observed binding of the LNK SH2 domain to phosphopeptides corresponding to these residues.

**Crystal Structure of the LNK SH2 domain in complex with the EPOR pY454 motif.** To understand how the EPOR pY454 peptide, which contains a leucine at the +1 position, could bind the LNK SH2 domain with high affinity, we determined the complex structure to 2.35 Å resolution (Supplementary Table 1). Overall, the structure is very similar to the LNK SH2/JAK2 pY813 structure with an RMSD of 1.0 Å over 111 residues (DALI[34]) (Fig. 4a and Supplementary Fig. 5). The LNK SH2 domain accommodates the leucine of the EPOR pY454 peptide at the key +1 position in a subtly different manner to the glutamate of the JAK2 pY813 peptide. The aliphatic portions of both the +1 glutamate and leucine sit on a hydrophobic surface of LNK, however the glutamate sidechain bends so that the carboxylate faces away and forms a salt bridge with Lys 384 (Fig. 4b, c). The side chain of the EPOR pY454 +1 leucine is accommodated by a shift in the BG loop of LNK by ~5 Å, although the exact positioning may be influenced by an adjacent crystal contact. In concordance with their similar binding modes, the surface area buried by each peptide is similar (~580 Å).

**Mutations found in human LNK impair target binding.** Several substitutions in LNK have been linked with diseases including haematological cancers[11,35], autoimmune disorders[24,36] and heart disease[21,23,37]. Most of these occur within the PH domain however substitutions located within the SH2 domain have also been

**Table 2 Peptide sequences and IC$_{50}$ values from phosphopeptide competition assay.**

| Protein | Phosphotyrosine | −7 | −6 | −5 | −4 | −3 | −2 | −1 | pY | +1 | +2 | +3 | +4 | +5 | +6 | +7 | IC$_{50}$ (nM)[a] |
| --- | --- | --- | --- | --- | --- | --- | --- | --- | --- | --- | --- | --- | --- | --- | --- | --- | --- |
| JAK2 | pY813 | | | | F | T | P | D | **pY** | E | L | L | T | E | N | D | 138 |
| | pY1007/pY1008 | A | I | E | T | D | K | E | **pY** | Y | K | V | K | E | P | G | >10,000 |
| | pY1007 | A | I | E | T | D | K | E | **pY** | Y | K | V | K | E | P | G | >10,000 |
| | pY1008 | I | E | T | D | K | E | Y | **pY** | K | V | K | E | P | G | | >10,000 |
| JAK3 | pY785 | | | | L | S | S | D | **pY** | E | L | K | S | D | | T | 305 |
| IRK | pY1445/pY1446 | | | | | E | T | T | **pY** | R | R | L | G | G | K | G | >10,000 |
| MPL | pY521 | | | | F | P | A | D | **pY** | L | R | D | R | H | A | W | >10,000 |
| | pY552 | | | | V | L | G | H | **pY** | R | R | L | T | A | A | L | >10,000 |
| | pY591 | | | | A | Q | D | Q | **pY** | L | R | L | Q | P | S | C | >10,000 |
| | pY626 | | | | A | N | H | M | **pY** | W | P | Q | S | Y | W | Q | 4887 |
| | pY631 | | | | L | P | L | S | **pY** | Q | Q | G | P | | | | >10,000 |
| EPOR | pY309 | | | | Q | L | W | S | **pY** | L | N | L | C | L | W | W | >10,000 |
| | pY368 | | | | A | S | D | L | **pY** | T | V | L | D | K | W | L | 1121 |
| | pY426 | | | | A | H | F | T | **pY** | L | L | L | D | P | S | S | 1557 |
| | pY454 | | | | P | D | L | E | **pY** | L | Y | P | Y | V | N | | 661 |
| | pY485 | | | | S | Q | G | K | **pY** | S | N | Q | Y | E | N | S | >10,000 |
| FLT3 | pY572 | | Y | K | K | E | F | R | **pY** | E | S | Q | L | Q | M | | >10,000 |
| | pY591 | | S | D | N | | Y | F | **pY** | V | D | F | R | E | Y | | 1250 |

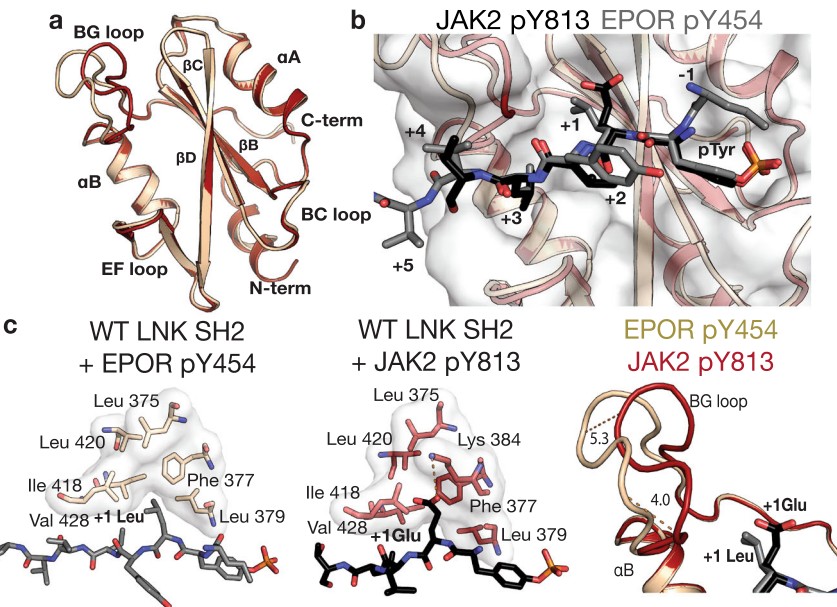

**Fig. 4 Comparison of the LNK SH2 domain bound to JAK2 pY813 and EPOR pY454 phosphopeptides. a** Alignment of the backbone of the WT LNK SH2 domain from the JAK2 pY813 bound structure (red) and the EPOR pY454 bound structure (purple) with secondary structural features indicated. **b** The pY813 JAK2 (black) and EPOR pY454 (white) peptides bound to the SH2 domain show a high degree of similarity. **c** The +1 Leu of the EPOR pY454 peptide is positioned in a hydrophobic surface on the LNK SH2 domain (left). The +1 Glu of the JAK2 pY813 peptide is positioned over the same surface, however the carboxylate also forms a salt bridge interaction with Lys 384 of LNK (middle). The LNK SH2 BG loop of the EPOR pY454 bound structure is displaced by approximately 5 Å in comparison to the JAK2 pY813 bound structure (red) (right).

described in patients with a variety of diseases including the myeloproliferative neoplasms, a group of disease characterised by excessive JAK-STAT signalling resulting in hyperplasia of specific mature blood lineages[11,13,20,38,39] and idiopathic erythrocytosis[40,41] (Fig. 5a). We investigated three such mutations which clustered around the phosphotyrosine binding site in the LNK SH2 domain, V402M, R415C and R415H (V374M R387C/H in mouse respectively) (Supplementary Fig. 6).

Using mouse SH2 domain equivalents of these variants we investigated whether the incorporation of these mutations affected protein stability by measuring their melting point ($T_m$). The apo WT LNK SH2 domain had a Tm of approximately 60 °C whereas all three mutants had a lower Tm of approximately 55 °C (Fig. 5b). Upon addition of phosphopeptide, the melting point of the WT and both Arg 387 mutants increased, consistent with the SH2 domain being stabilised by the binding of peptide, whereas the V374M mutant did not. This suggests that R387C and R387H retain the ability to bind phosphorylated sequences whilst V374M does not.

In order to investigate this further, direct binding of the WT and mutant LNK SH2 domains to the JAK2 pY813 sequence was examined by SPR. All three variant LNK SH2 domains showed compromised binding to the pY813 peptide. As shown in Fig. 5c, the R387C and R387H mutants displayed a 3-fold and 10-fold loss in affinity relative to the WT SH2 domain whilst V374M was too weak to quantify (Supplementary Fig. 7). Binding of the LNK SH2 domains was also compared to that of SH2B, which has also been shown to bind JAK2 pY813[2]. The SH2B SH2 domain binds pY813 with an affinity of ~100 nM (Fig. 5c), however the stability of the SH2B complex was significantly lower than for LNK as evidenced by much faster dissociation compared to the LNK/phosphopeptide interaction, which is characterised by a slow off-rate (Supplementary Fig. 7). Interestingly, the defect in the two R387 mutants appears to mostly be due to a compromised on-rate (Supplementary Fig. 7). Together these data indicate that

these three mutations surrounding the phosphotyrosine binding pocket of the LNK SH2 domain interfere with substrate binding. This is further supported by thermal shift data indicating compromised binding of phosphotyrosine (Supplementary Fig. 8).

To determine whether these mutations lead to an effect on the regulation of signalling, the three *H. sapiens* LNK SH2 domain mutants were expressed as full-length proteins in HEK293 cells and their ability to regulate signalling downstream of IFN-γ treatment was compared to WT by luciferase assay (Fig. 5d). All three mutants had a significantly reduced capacity to regulate signalling, with the V374M mutant being the most compromised, consistent with biochemical data indicating disruption of phosphotyrosine binding.

## Discussion

The data presented here reveals key determinants for LNK SH2 domain ligand recognition and identifies three high-affinity phosphotyrosine-motifs in JAK2, JAK3 and EPOR. Our structural data shows that both the JAK2 and EPOR sites bind LNK in a canonical fashion, similarly to SH2B but in contrast to APS. Although the +1 residues in the EPOR and JAK2 peptides differ, glutamate vs. leucine respectively, their binding can be accommodated by the same hydrophobic residues in the +1 pocket. Together with the structural data, our biochemical analysis indicates that residues in the +1, +3 (and to a lesser extent, +5) positions, relative to pTyr, confer specificity for the LNK SH2 domain and contribute to high-affinity binding.

In addition to the high-affinity JAK2 pY813, JAK3 pY785 and EPOR pY454 sites, we identified moderate-affinity motifs in EPOR (pY426 and pY368), FLT3 (pY591 and pY919) and C-KIT pY568 as potential binding sites for the LNK SH2 domain. These results are in agreement with previous findings[4,5,10,16,18,19]. Surprisingly, we did not find any binding sites in the TPO receptor. This may indicate that LNK interacts with the TPO signalling cascade only

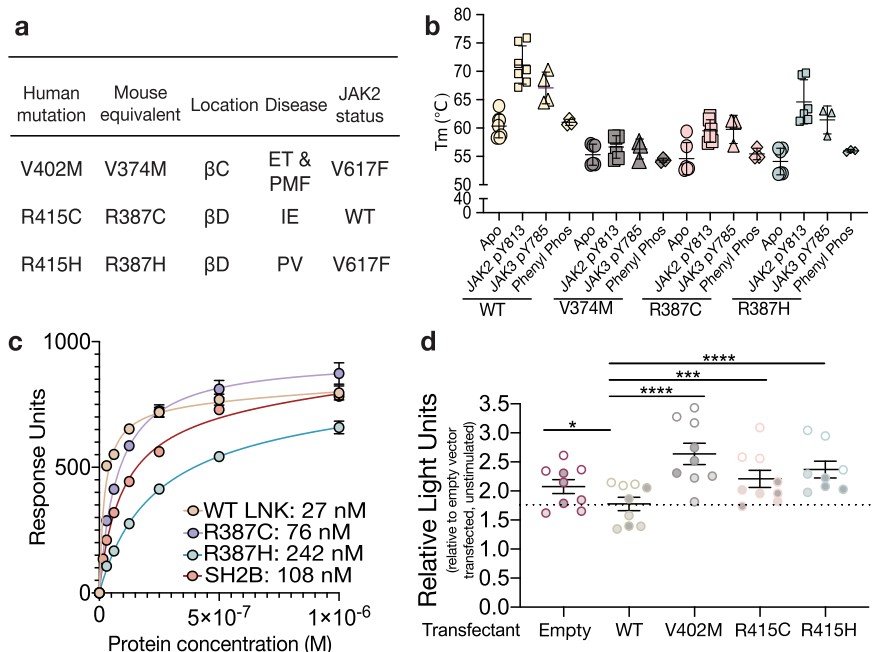

**Fig. 5 Characterisation of three LNK SH2 domain mutations. a** Three variants investigated in this study, their location within the LNK SH2 domain, the disease they were identified in (PV polycythemia vera, ET essential thrombocythemia, PMF primary myelofibrosis, IE idiopathic erythrocytosis) and the JAK2 status of the patient. **b** Melting temperature (Tm) in °C for apo LNK SH2 domains and with JAK2 pY813 and JAK3 pY785 peptides and the pTyr mimetic, phenyl phosphate (Phenyl phos). Data are displayed as the mean ± SD of technical replicates from n = 3 independent experiments. **c** Fitted curves for WT, R387C and R387H LNK SH2 domain, and SH2B SH2 domain affinity for JAK2 pY813. Data are displayed as mean ± SD from n = 5 independent experiments for WT and R387C LNK SH2 domains and n = 4 independent experiments for LNK R387H and n = 3 independent experiments for SH2B SH2 domains. $K_D$ for each SH2 domain is indicated in nM. V374M did not bind. **d** Relative GAS-Firefly luciferase activity after co-transfection of HEK293 with WT, V402M (V374M), R415C (R387C) or R415H (R387H) full-length human LNK mutants and treatment with 50ng/ml rhIFN-γ. Data are displayed as mean ± SEM from n = 3 independent experiments where data were pooled from three experiments with each data point representing the mean of triplicate replicates of a single transfection normalised to the mean RLU (relative light units) of unstimulated empty vector control in each experiment. Statistical analysis was performed using two-sided pairwise multiple comparison with Tukey's adjustment under estimated marginal means (emmeans) function based on linear mixed-effect models using each individual experiment as a block. Significance is indicated with asterisks: *$p < 0.05$, ***$p < 0.001$, ****$p < 0.0001$ (empty control $p = 0.05$, V402M $p < 0.0001$, R415C $p = 0.0006$ and R415H $p < 0.001$).

via JAK2, although it is possible that other domains in the protein may modulate an interaction with this receptor. To date, we have been unable to purify the full-length protein.

The documentation of LNK mutations in patients with MPNs has generated interest in understanding the role of LNK in haematological diseases. Although mutations in LNK are not commonly found as drivers of MPNs, evidence has emerged that there is an overrepresentation of LNK mutations in blast-phase MPN, suggesting a role in leukemic transformation[13,14]. Here we investigated three single amino acid substitutions located within the LNK SH2 domain that had been identified in MPNs or the closely related disease, idiopathic erythrocytosis; V402M, R415C, and R415H (V374 and R397 in mouse)[11,15,39,40,42]. Of these, V374M had the most severe effect on peptide binding. V374 is positioned directly underneath the pTyr ring, and pTyr binding is presumably blocked by the larger methionine side chain. The effect of the R387C/H LNK SH2 domain variants were more subtle. We observed a slight loss in thermal stability and affinity with these two mutants but they were still able to bind phosphorylated motifs to some extent. We hypothesise that the loss of the positive charge leads to a slight destabilisation of the BC loop (which forms the base of the pTyr binding pocket) due to disruption of an R369-E372-R387 hydrogen-bonding network. Disruption of the Arg387 could manifest as both a slightly reduced thermal stability and also a slower on-rate (the domain is not "primed" to bind pTyr). These mutations also effected LNK function in cells as in vitro analyses performed with the full-length human protein revealed that all three mutants had a

reduced capacity to negatively regulate cytokine signalling, with V402M being the most significantly compromised. These findings emphasise differences in mutation type, and highlight how identification of particular mutations in patients may be useful in assessing risk of transformation.

Taken together, these results provide a detailed picture of how the LNK SH2 domain interacts with its ligands and coupled with future studies to determine the full repertoire of LNK-interactors in cells, will allow an understanding of how defects in LNK contribute to myeloproliferative, and other haematological diseases.

## Methods

**Expression and purification of LNK and SH2B SH2 domains**. DNA encoding the mouse LNK SH2 domain (residues 324–446) and an N-terminal NusA[43] fusion separated by a TEV cleavage site was cloned (primers in Supplementary Table 2) into an pET-50b(+) vector using BamH1 and Not1, and transformed into tuner (DE3) (Novagen), BL21(DE3), or C43 *E. coli* cells and expression was induced by addition of 1 mM IPTG at 18 °C overnight. Cells were collected by centrifugation and frozen at −30 °C. Cells from 1 L culture were resuspended in 40 mL lysis buffer (20 mM Tris (pH 8.0), 10 mM imidazole (pH 8.0), 300 mM NaCl, 2 mM TCEP, 5 mM phenyl phosphate, 1 U DNAse, 1 mM PMSF, and 20 mg lysozyme) containing EDTA-free protease inhibitor cocktail (SigmaAldrich) and lysed by sonication. Lysate was clarified by spinning cells for 10 min at 20,000 × *g* before loading supernatant onto complete ™ His-Tag purification resin (Merck). Bound proteins were washed with 20 mM tris (pH 8.0), 10 mM imidazole (pH 8.0) and 300 mM NaCl, 5 mM phenyl phosphate followed by 20 mM tris (pH 8.0), 30 mM imidazole (pH 8.0) and 300 mM NaCl, 2 mM TCEP 5 mM Phenyl Phosphate. Protein was eluted in 20 mM tris (pH 8.0), 250 mM imidazole (pH 8.0) and 300 mM NaCl, 5 mM Phenyl Phosphate and 2 mM TCEP. Eluate was then cleaved with TEV protease overnight at 4 °C and subsequently purified further by size exclusion

chromatography (Superdex 200 26/600 from GE healthcare) in TBS, 2 mM TCEP and 5 mM Phenyl Phosphate.

DNA encoding the mouse SH2B SH2 domain and an N-terminal GST fusion separated by a TEV cleavage site was transformed into BL21(DE3) *E. coli* cells and expression was induced by addition of 1 mM IPTG at 18 °C overnight. Cells from 1 L culture were resuspended in 40 mL lysis buffer (Phosphate buffered saline (PBS), 5 mM DTT, 1 U DNASe, 1 mM PMSF, and 20 mg lysozyme) and lysed by sonication. Lysate was clarified by spinning cells for 10 min at 20,000 × *g* before loading supernatant onto Glutathione agarose resin purification resin (UBPbio). Bound proteins were washed with PBS, 5 mM DTT. Protein was then cleaved with TEV protease overnight at 4 °C and cleaved SH2B SH2 domain was subsequently purified further by size exclusion chromatography (Superdex 200 10/300 from GE healthcare) in TBS, 5 mM DTT.

**Thermostability assays**. For peptide and phenyl phosphate experiments, proteins were desalted into 100 mM NaCl, 20 mM Tris (pH 8.0), 2 mM TCEP buffer and diluted to 100 µM. Where peptides were used, a five-fold molar excess of peptide was added to each sample, in phenyl phosphate conditions concentration was 8 mM. For experiments were phosphotyrosine, proteins were diluted to 30 µM in PBS and in phosphate conditions, concentration was 10 mM. Ten mcroliter of each sample was transferred into a capillary and measured from 35 to 95 °C using a Tycho N6T (Nanotemper). Data were analyzed in Prism.

**SPR competition assays**. SPR competition assays were performed on either a Biacore 4000 or 200 (GE Healthcare) in 10 mM HEPES (pH 7.4), 150 mM NaCl, 3.4 mM EDTA, 0.005% Tween 20 using a streptavidin coated chip and were regenerated in 50 mM NaOH, 1 M NaCl. A biotinylated peptide representing the IL6ST pY757 sequence was immobilised to the chip by passing over 1 µg/mL of peptide dissolved in 10 mM HEPES pH 7.4, 150 mM NaCl, 3.4 mM EDA, 0.005% Tween 20. 0.1–0.5 µM of WT LNK SH2 domain was pre-incubated with 2, 1, 0.5, 0.25, 0.125 or 0 mM phosphopeptides before being flowed over the chip for 240–720 seconds at 10–30 µL/min. Data was analysed using Prism, the response of LNK in the presence of the peptide was normalised to an LNK only control and then fitted as an IC$_{50}$ curve via non-linear regression.

**SPR direct binding assay**. Direct binding experiments were performed on either a Biacore 4000 or 8000 (GE Healthcare) in 10 mM HEPES (pH 7.4), 150 mM NaCl, 3.4 mM EDA, 0.005% Tween 20 and were regenerated in 50 mM NaOH, 1 M NaCl. LNK and SH2B SH2 domains were flowed over a streptavidin coated chip for 420 s at 30 µL/min with immobilised biotinylated JAK2 pY813 bound to determine binding kinetics and affinity. A reference flow cell was included by passing buffer without protein over a single lane and the sensorgrams from the reference cell were subtracted from the experimental flow cell analyses. Data were subsequently plotted in Prism.

**Crystallography**. All LNK constructs were buffer exchanged into low salt buffer (20 mM Tris (pH 8.0), 2 mM TCEP and 100 mM NaCl) and crystal trays were set up with 5 mg/mL of protein and a 2-fold molar excess of peptide using vapour diffusion sitting drop experiments at the collaborative crystallisation centre, CSIRO (JAK2 pY813) or hanging drops set up in house (EPOR pY454). The WT LNK SH2 domain with the JAK2 pY813 peptide crystalized in 0.2 M ammonium chloride, 20% w/v PEG 3350. The WT LNK SH2 domain with the EPOR pY454 peptide crystalised in 20% w/v PEG 8000, 0.05 magnesium acetate, 0.1 M Tris (pH 8.5). All crystals were cryo-protected in paratone and immediately snap frozen in liquid nitrogen. Data was collected at the MX2 beamline at the Australian Synchrotron. Data reduction, scaling and integration was performed using XDS[44]. Crystal structures of the LNK SH2 domain were solved by molecular replacement (search model PDB ID: 2HDX for WT LNK SH2/JAK2 pY813 structure, and WT LNK SH2 for WT LNK SH2/EPOR pY454) using Phaser as implemented in PHENIX[45]. All structurers were refined using PHENIX and model building was performed in COOT[46]. Structures were visualised using Pymol[47].

**Luciferase assay**. V402M, R415C and R415H mutations were introduced into the open reading frame (ORF) of pReceiver-M12-*SH2B3* (Genecopoeia, Rockville, MD, USA) construct by site-directed mutagenesis using PfuTurbo DNA polymerase (Santa Clara, CA, USA) and 5% DMSO to resolve secondary structure due to GC-rich sequence in *SH2B3* ORF (primers in Supplementary Table 2).

Twenty-four-well plate cultures of HEK293 cells at ~70% confluency in were transfected with 100 ng of either empty vector or full-length human LNK encoding pReceiver-M12 constructs, 145 ng of GAS-Firefly luciferase reporter construct (a gift from Vicki Athanasopoulos) and 5 ng of pRL-Renilla luciferase control construct using Lipofectamine 2000 transfection reagent (Invitrogen, Carlsbad, CA, USA). Twenty-four hours of post-transfection, HEK293 cells were stimulated with 50ng/mL recombinant human IFN-γ (Peprotech, Rock Hill, NJ, USA) for 24 h. Firefly and Renilla luciferase activity were measured using a Luc-Pair Duo-Luciferase Assay Kit (Genecopoeia, Rockville, MD, USA) and a VICTOR Nivo multimode plate reader (PerkinElmer, Waltham, MA, USA). Relative light unit (RLU) was calculated as the mean of ratios between the GAS-Firefly luciferase luminescence and Renilla luciferase luminescence, and normalised to the mean RLU of the unstimulated empty vector-transfected samples to calculate relative GAS activity. Statistical analysis was performed using the lmerTest package in R.

**Reporting summary**. Further information on research design is available in the Nature Research Reporting Summary linked to this article.

## Data availability
The data that support this study are available from the corresponding author upon reasonable request. Atomic coordinates for WT LNK SH2 domain with the JAK2 pY813 and EPOR pY454 phosphopeptides have been deposited in the Protein Data Bank with the accession numbers PDB 7R8W and PDB 7R8X, respectively. Source data are provided with this paper.

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

## Acknowledgements

We sincerely thank the late Prof. Tony Pawson, discoverer of the SH2 domain, who first approached and inspired us to study this protein. We are very grateful to Kazuya Machida for the details of their lnk construct. This work was supported by the National Health and Medical Research Council (NHMRC) Australia (Project grant no. 1122999, Program grant no. 1113577), an NHMRC IRIISS Grant 9000220, and a Victorian State Government Operational Infrastructure Scheme grant. J.J.B. is supported by an NHMRC fellowship. RM was supported by an Australian Postgraduate Award. This research was undertaken in part using the MX2 beamline at the Australian Synchrotron, part of ANSTO, and made use of the Australian Cancer Research Foundation (ACRF) Eiger detector[48]. Crystallization trials were performed at CSIRO collaborative crystallization centre (C3).

## Author contributions

R.M., Y.Z., J.M.M., A.L. Carried out experiments. R.M., Y.Z., J.M.M., A.L., J.I.E., C.G.V., N.J.K., J.J.B. designed experiments and analysed and interpreted data. R.M., N.J.K., collected and analysed crystallographic data. R.M., N.J.K., J.J.B. wrote the manuscript. R.M., Y.Z., J.I.E., J.M.M., N.J.K., and J.J.B. revised the manuscript. J.J.B. Conceived project and designed the study.

## Competing interests

The authors declare no competing interests.
