## [Peer Review File · Nature Communications]

Structural and functional analysis of target recognition by the lymphocyte adaptor protein LNKReviewers' Comments:

Reviewer #1:

Remarks to the Author:

In this paper, Morris et al characterize LNK SH2 domain interactions relevant to thrombopoietin (TPO) and erythropoietin (EPO) signaling. Mutations in the SH2 domain are known to cause disease and there is clear rationale for their structural and biochemical investigation of LNK, specifically that related proteins APS and SH2B have previously been shown to bind targets in distinct ways (e.g., APS dimerizes and SH2B does not). The authors determine two crystal structures of LNK-peptide complexes and find that LNK SH2 is more similar to that of SH2B. In addition, the authors make a number of amino acid substitutions in target peptides and measure resulting IC50 values (using surface plasmon resonance). They also determine IC50 values for various pY-containing peptide sequences in putative binding partners, including c-KIT, c-FMS, FLT3, and PDGFR. Finally, the authors investigate 3 of the disease-causing mutations in LNK SH2, with respect to thermal stability, resulting IC50 values with selected peptides, and (very interestingly) signaling in cell culture, using a luciferase assay.

Overall, the authors have done a nice job of describing their Results and discussing the significance of their work, with some minor editing comments (as suggested below, most notably in their first results paragraph which does not properly describe why the mouse LNK SH2 domain was chosen or the expression/purification/crystallization). This is a very nice example of rigorous protein biochemistry and structural biology that does a good job of describing peptide recognition by SH2 domains not previously characterized. This study also highlights how important it is to look at specific interactions, even if related proteins have previously been characterized. I do not think these authors need to do additional experiments, although it would be nice to see if the disease-causing mutations affect a larger pool of their putative substrates (and not just the one they test). Even just adding data for EPOR pY454 would be nice, considering the proposed "alternative" binding at the +1 position. There are several typos and references that need to be inserted (as described below) and the authors should do a close read through of their paper prior to resubmission. But overall, very nice job!

Minor comments:

Introduction:

- Your introduction would be stronger with a few more descriptive sentences. Perhaps instead of starting with LNK, you could start with why we care about TPO and EPO signaling? Then, you could introduce LNK and the other SH2-containing proteins. Along these lines, you should cite the original Lnk cloning paper (Huang X et al, PNAS, 1995; <https://www.pnas.org/content/pnas/92/25/11618.full.pdf>) and mention what "Lnk" stands for/how it was named. Additional acronyms (e.g., JAK) should also be defined in the text and not just on the first page, but perhaps that is personal preference. The rest of the Introduction is nicely written, so it is really only these first couple of sentences that do not flow as well.

- Typo, second paragraph - reference not properly inserted "(Hurtado et al. 2011)."

- Hypothesis: LNK SH2's ability to interact with its substrates affects regulation of downstream signaling (?) - this could be made more clear in the Introduction. The introduction focuses a bit much on the full-length protein, but the work is mostly on the isolated SH2 domain and the authors very nicely characterize this with respect to target binding and signaling.

- "Without a comprehensive overview of the proteins and pathways that LNK negatively regulates, our knowledge of its physiological roles is incomplete (Page 3)." -> Does this paper get to this? Wouldn't this require proteomics? Perhaps this statement should be moved to the Discussion as a future direction/idea. In the Introduction, it is a bit misleading considering the authors are only looking at

SH2 domain-mediated interactions.

- "Studies on SH2 domain point mutations identified in patients with myeloproliferative neoplasms showed a decrease in affinity for ligand and decreased ability to regulate signaling in vitro (Page 4)." This is a very important conclusion in the paper and the authors should make clear in the Introduction that they conducted these studies using full-length protein in a cell-based assay.

Results:

- The results should start with a brief description of LNK expression and purification and crystallization. The authors make a point to say that there is only one other example of LNK expression (of any part of the protein), so were there issues in purifying the SH2 domain? Is this why they used the mouse construct? How did they verify that it was monomeric and well-behaved? SDS-PAGE? SEC? Many of these details are in the Methods, but there needs to be some abbreviated description in the Results as well, since this protein has not been purified previously and this is also a major result.

- Why was the mouse protein used and not the human sequence? There is no justification provided.

- Supplementary Figures 2 and 3 are out of order, based on the text.

- Typo (page 5) "disulphide" should be disulfide.

- Missing reference insertion, page 6, "(Hu and Hubbard 2006)"

- Personal preference, but I would prefer to see structural alignment of two proteins with ~70% identity be done using main chain atoms only.

- The numbering of residues (+1, +2, etc.) should be described much earlier in the text, perhaps in the introduction.

- Typo (page 7), "While the IC50 [of] the WT pY813..."

- The authors should be more descriptive about their binding assay results. For example, instead of saying that an interaction is "substantially" weaker, the authors could state (based on their Table 2 data) that "substitution of +1 Glu to Ala resulted in an ~7-fold reduction in IC50, whereas +1 Glu to Arg was >80-fold weaker." Note: They do this nicely later on in the paper.

- Missing reference insertion, page 8, "(Holm and Sander 1995)"

- Why is "Figure 4" in bold on page 8 when the figures are not bolded elsewhere?

- Typo, page 8, missing period at end of paragraph, "... the surface area buried by each peptide is similar (~580 Angstrom)." (Also, there should be a reference here for how buried surface area was determined)

- The final Results section, "Mutations found in human LNK impairs target binding" is very nice and the most human health relevant data in this paper. It is written almost as an afterthought, however. (1) Why weren't more of the peptides tested with the mutant proteins? (2) Should the luciferase assay be a separate Results section, considering how critical it is to the authors' main conclusions?

Discussion:

- "... and moves us closer to a comprehensive list of the proteins with which LNK can associate." The statements in this paper alluding to the full interactome of LNK are a bit overstated, considering the authors only look at the SH2 domain. Here, (and earlier) it should perhaps be edited to focus on describing interactions at the SH2 domain alone.

Figures:

Figure 1b: The peptide positions in the middle figure should be labeled, "+1 Glu," etc (note: this is nicely done in Figs. 1c and 1d). In the right hand figure, residue numbers should be added. Also, the authors should make sure that amino acid labels are readable (e.g., the "+1" in the middle figure).

Figure 1c: The authors should make sure all amino acid labels are clearly readable, e.g., "+3 Leu," "Ser 366," etc.

Figure 5: The figure legend is mislabeled (it reads "Figure 6"). Also the authors should describe (in the legend or on the figure) what "RH" and "RC" stand for in Fig. 5c, and not leave it to the reader to decipher.

Reviewer #2:

Remarks to the Author:

The study by Morris et al. examines the adaptor protein LNK, a member of the SH2B family of three related SH2-containing signaling proteins (SH2B1–3). Crystal structures of the SH2 domain of family members SH2-B (SH2B1) and APS (SH2B2) have been reported previously. In this study, the authors determine the co-crystal structures of the LNK (SH2B3) SH2 domain in complex with two phosphopeptides, from JAK2 (pY813) and from EpoR (pY454). These structures reveal that the LNK SH2 domain is conventionally constructed, similar to that of SH2-B (but dissimilar to that of APS), and bind phosphopeptides in the conventional manner. There are subtle differences between the modes of binding of pY813 to the LNK SH2 domain versus the SH2-B SH2 domain. The co-crystal structure with pY454 shows how LNK can recognize leucine as well as arginine at the P+1 position of the phosphopeptide (not an easy feat).

Using an SPR competition assay, the authors measured the binding of the LNK SH2 domain to various phosphopeptides proposed to interact with LNK. These results will be useful in evaluating potential binding sites in vivo.

Various genetic data indicate that LNK serves as a negative regulator of cytokine signaling through an unknown mechanism(s) and may be a contributing factor in some forms of myeloproliferative neoplasms in humans. Several loss-of-function mutations are present in the SH2 domain of LNK as well as in the PH domain. The authors purified three such mutant forms of the LNK SH2 domain and characterized their stability and ability to bind phosphopeptides. All three mutants exhibited a lower melting point than the wild-type LNK SH2 domain, indicative of lower stability, and one of the mutants lost the ability to bind phosphopeptides. The phosphopeptide binding affinities were quantified by SPR analyses. These mutations were also tested in full-length LNK for their ability to regulate IFN γ signaling, the results of which were consistent with the biochemical data showing impaired phosphopeptide binding.

Overall, this study will be of interest to those in the cytokine signaling field. Outside the scope of this study, but of great interest, is the mechanism by which LNK serves as a negative regulator of cytokine signaling, especially since SH2-B appears (generally) to be a positive regulator of signaling, particularly in tumor progression. However, one doable experiment that would be informative in this context is a head-to-head measurement of the binding affinities of JAK2 pY813 to the LNK and SH2-B SH2 domains, since both SH2 domains are thought to bind this site in vivo.

Minor issue: There are problems in the text in referencing parts of Figure 5 and non-existent (but labeled as such) Figure 6.

Response to Reviewers (Morris *et al.*)

We sincerely thank the reviewers for their very supportive and helpful comments and for their valuable time that they have spent helping to improve our manuscript. We have addressed the reviewers comments as outlined below:

Reviewer 1.

"Your introduction would be stronger with a few more descriptive sentences. Perhaps instead of starting with LNK, you could start with why we care about TPO and EPO signaling? Then, you could introduce LNK and the other SH2-containing proteins. Along these lines, you should cite the original Lnk cloning paper (Huang X et al, PNAS, 1995; and mention what "Lnk" stands for/how it was named. Additional acronyms (e.g., JAK) should also be defined in the text and not just on the first page, but perhaps that is personal preference. The rest of the Introduction is nicely written, so it is really only these first couple of sentences that do not flow as well."

Response: We have cited the Huang et al paper, defined JAK and LNK and modified the beginning of the introduction to:

Thrombopoietin (TPO) and erythropoietin (EPO) are lineage-dominant cytokines that bind specific receptors on the surface of target cells and induce activation of the janus kinase (JAK)-Signal transduce and activation of transcription (STAT) pathway. TPO is an essential modulator of megakaryopoiesis, platelet production and stem cell quiescence, whereas EPO required for the regulation of erythropoiesis. The Lymphocyte adaptor protein (LNK¹ (also SH2B3)) is a member of the SH2 domain containing adaptor family of proteins, which also comprises APS (SH2B2) and SH2B (SH2B1)^{2,3} and negatively regulates both EPO and TPO signalling^{4,5} via its interaction with JAK2⁶

"Typo, second paragraph - reference not properly inserted "(Hurtado et al. 2011)."

Response: Reference inserted

"Hypothesis: LNK SH2's ability to interact with its substrates affects regulation of downstream signaling (?) - this could be made more clear in the Introduction. The introduction focuses a bit much on the full-length protein, but the work is mostly on the isolated SH2 domain and the authors very nicely characterize this with respect to target binding and signaling."

Response: We have adjusted several sections in the introduction to make it clear we specifically refer to the LNK SH2 domain and state that it is the substrate recognition domain. Changes in red in the accompanying text.

"Without a comprehensive overview of the proteins and pathways that LNK negatively regulates, our knowledge of its physiological roles is incomplete (Page 3)." -> Does this paper get to this? Wouldn't this require proteomics? Perhaps this statement should be moved to the Discussion as a future direction/idea. In the Introduction, it is a bit misleading considering the authors are only looking at SH2 domain-mediated interactions."

Response: We replaced this sentence with "Characterisation of the substrate recognition domain of LNK, the SH2 domain, would therefore be a step towards a comprehensive overview of the proteins and pathways that LNK negatively regulates." to make it clear that this is a more targeted approach. The reviewer is certainly correct that sophisticated proteomics would be required for a full

characterisation of the targets of Lnk action in cells.

“The results should start with a brief description of LNK expression and purification and crystallization. The authors make a point to say that there is only one other example of LNK expression (of any part of the protein), so were there issues in purifying the SH2 domain? Is this why they used the mouse construct? How did they verify that it was monomeric and well-behaved? SDS-PAGE? SEC? Many of these details are in the Methods, but there needs to be some abbreviated description in the Results as well, since this protein has not been purified previously and this is also a major result.”

We thank the reviewer for this suggestion as in fact getting soluble, active Lnk protein was the major roadblock for us at the beginning of this study. We tried many different expression systems to produce soluble and folded LNK protein (mouse and human) and were unsuccessful until we tried an identical construct to that used by Machida et al (as a NusA fusion). This made all the difference for us and we are indebted to Machida for sending us the full details of his clone. We ensured the protein was monomeric by SEC and that it was correctly folded by thermal denaturation experiments and pTyr binding. We have added the following to the results section to describe this:

Response: **The following sentences added:**

We attempted a number of different bacterial and baculovirus expression systems to produce active SH2 domain from both human and mouse LNK including. Whilst GST and His6-based fusion systems were unsuccessful, a NusA fusion of the *M. musculus* LNK SH2 domain (using the domain boundaries from Machida et al²⁸) yielded folded, monomeric protein once the fusion tag was removed as determined by size exclusion chromatography and thermal shift assay.

“Why was the mouse protein used and not the human sequence? There is no justification provided”

Response: **We attempted to produce the human protein, however even using the same NusA fusion system were unable to produce soluble, folded protein.**

Supplementary Figures 2 and 3 are out of order, based on the text.

Response: **Order of supplementary figures 2 and 3 have been changed**

“Typo (page 5) “disulphide” should be disulfide.”

Response: **Changed to disulfide**

“Missing reference insertion, page 6, “(Hu and Hubbard 2006)”

Response: **Reference inserted**

“Personal preference, but I would prefer to see structural alignment of two proteins with ~70% identity be done using main chain atoms only.”

Response: **Changed to alignment of Ca atoms and text changed to “1.26 Å over 106 atoms”**

“The numbering of residues (+1, +2, etc.) should be described much earlier in the text, perhaps in the introduction.”

Response: The following sentence was added on line 111:
(designated below as +1,+2.... and -1, -2... relative to pTyr)

“Typo (page 7), “While the IC50 [of] the WT pY813...”

Response: **The word “of” added to the sentence**

“The authors should be more descriptive about their binding assay results. For example, instead of saying that an interaction is “substantially” weaker, the authors could state (based on their Table 2

data) that "substitution of +1 Glu to Ala resulted in an ~7-fold reduction in IC50, whereas +1 Glu to Arg was >80-fold weaker." Note: They do this nicely later on in the paper."

Response: Changed this sentence to be more descriptive. Text changed to:

In particular, "mutation of the +1 Glu to Arg and the +3 Leu to Phe resulted in a >80-fold and ~25-fold decrease in affinity respectively (Table 2)".

"Missing reference insertion, page 8, "(Holm and Sander 1995)"

Response: Reference now inserted correctly

"Why is "Figure 4" in bold on page 8 when the figures are not bolded elsewhere?"

Response: Figure 4 was Unbolded

"Typo, page 8, missing period at end of paragraph, "... the surface area buried by each peptide is similar (~580 Angstrom)." (Also, there should be a reference here for how buried surface area was determined)"

Response: Full stop added to this sentence.

"The final Results section, "Mutations found in human LNK impairs target binding" is very nice and the most human health relevant data in this paper. It is written almost as an afterthought, however. (1) Why weren't more of the peptides tested with the mutant proteins? (2) Should the luciferase assay be a separate Results section, considering how critical it is to the authors' main conclusions?"

Response: As requested by the reviewer, we aimed to perform this experiment until a COVID-induced lockdown prevented us from doing these experiments. We were however able to perform a thermostability experiment with the mutant SH2 domains and phosphotyrosine. Given these mutants cluster around the phosphotyrosine binding pocket, we would expect to effect binding of all phosphopeptides. This is supported by our experiment which shows compromised binding of phosphotyrosine for all three mutants relative to the WT. This data was added as a supplementary Figure (S8) and the following text was added:

"Together these data indicate that these three mutations surrounding the phosphotyrosine binding pocket of the LNK SH2 domain interfere with substrate binding. This is further supported by thermal shift data indicating compromised binding of phosphotyrosine alone (Supplementary Figure 8)."

"... and moves us closer to a comprehensive list of the proteins with which LNK can associate." The statements in this paper alluding to the full interactome of LNK are a bit overstated, considering the authors only look at the SH2 domain. Here, (and earlier) it should perhaps be edited to focus on describing interactions at the SH2 domain alone."

Response: We have removed this sentence and modified the final sentence of the discussion to describe future work. Text changed to:

Taken together, these results provide a detailed picture of how the LNK SH2 domain interacts with its ligands and coupled with future studies to determine the full repertoire of LNK-interactors in cells, will allow an understanding of how defects in LNK contribute to myeloproliferative, and other haematological diseases.

"The peptide positions in the middle figure should be labeled, "+1 Glu," etc (note: this is nicely done in Figs. 1c and 1d). In the right hand figure, residue numbers should be added. Also, the authors should make sure that amino acid labels are readable (e.g., the "+1" in the middle figure)."

Response: Figures were modified to ensure labels were legible and amino acids residue numbers

added.

"Figure 1c: The authors should make sure all amino acid labels are clearly readable, e.g., "+3 Leu," "Ser 366," etc."

Response: **Figure was modified to ensure all residues legible**

"Figure 5: The figure legend is mislabeled (it reads "Figure 6"). Also the authors should describe (in the legend or on the figure) what "RH" and "RC" stand for in Fig. 5c, and not leave it to the reader to decipher."

Response: **Figure legends was amended to read as Figure 5 and labels were modified to include mutant number instead of RC and RH.**

Reviewer 2.

"However, one doable experiment that would be informative in this context is a head-to-head measurement of the binding affinities of JAK2 pY813 to the LNK and SH2-B SH2 domains, since both SH2 domains are thought to bind this site in vivo"

Response: We thank the reviewer for this excellent suggestion as what we found was the affinity of LNK and SH2B for pY813 is similar however LNK forms a much more stable complex (vastly decreased off rate). **This experiment was conducted and the results were included in Figures 5c and Supplementary Figure 7. The following text was also added:**

"Binding of the LNK SH2 domain was also compared to that of the SH2B, which has also been shown to bind JAK2 pY813². The SH2B SH2 domain binds pY813 with an affinity of ~100nM (Figure 5c), however the stability of the SH2B complex was significantly lower than for LNK as evidenced by much faster dissociation compared to the LNK/phosphopeptide interaction, which is characterised by a slow off-rate (Supplementary figure 7)."

"There are problems in the text in referencing parts of Figure 5 and non-existent (but labeled as such) Figure 6"

Response: **References to figures fixed in text.**